# Hyperspectral Imaging and Machine Learning for Huanglongbing Detection on Leaf-Symptoms

**DOI:** 10.3390/plants14030451

**Published:** 2025-02-03

**Authors:** Ruihao Dong, Aya Shiraiwa, Katsuya Ichinose, Achara Pawasut, Kesaraporn Sreechun, Sumalee Mensin, Takefumi Hayashi

**Affiliations:** 1Faculty of Informatics, Kansai University, Osaka 569-1095, Japan; k975945@kansai-u.ac.jp; 2Electrical Engineering and Computer Science, Tottori University, Tottori 680-8552, Japan; shiraiwa@tottori-u.ac.jp; 3Independent Researcher, Tsukuba 300-1252, Japan; 4Royal Project Foundation, 910 Moo 3, T. Maehia, Muang, Chiang Mai 50200, Thailand; acrpwst@gmail.com (A.P.); ketcnx@gmail.com (K.S.); linly317@gmail.com (S.M.)

**Keywords:** citrus greening disease, *Citrus reticulata*, disease diagnosis, feature extraction, multivariate analysis

## Abstract

Huanglongbing is one of the most destructive diseases of citrus worldwide. Infected trees die due to the absence of practical cures. Thus, the removal of HLB-infected trees is one of the principal HLB managements for the regulation of disease spread. Here, we propose a non-destructive HLB detection method based on hyperspectral leaf reflectance. In total, 72 hyperspectral leaf images were collected in an HLB-invaded citrus orchard in Thailand and each image was visually distinguished into either any HLB symptom appearance (symptomatic) or no symptoms (asymptomatic) on the leaf. Principal component analysis was applied on the hyperspectral data and revealed 16 key wavelengths at red-edge to near-infrared regions (715, 718, 721, 724, 727, 730, 733, 736, 930, 933, 936, 939, 942, 945, 957, and 997 nm) that were characteristically differentiated in the symptomatic group. Seven models learnt on the spectral data at these 16 wavelengths were examined for the potential to separate these two image groups: random forest, decision tree, support vector machine, k-nearest neighbor, gradient boosting, logistic regression, linear discriminant. F1-score was employed to select the best-fit model to distinguish the two categories: random forest achieved the best score of 99.8%, followed by decision tree and k-nearest neighbor. The reliability of the visual grouping was evaluated by nearest neighbor matching and permutation test. These three models separated the two image categories as precisely as PCR results, indicating their potential as alternative tool instead of PCR.

## 1. Introduction

Huanglongbing (HLB), or citrus greening disease, is caused by the bacterium *Candidatus* Liberibacter asiaticus, which is transmitted between plants by the vector insect *Diaphorina citri* in Asian countries [1]. HLB is recognized as the most destructive disease of citrus worldwide. As no effective measures have been established yet [1], the current HLB management consists of three major measures: use of pathogen-free materials and sanitation of fields to plant trees, vector control, and removal of infected trees. The conventional detection of HLB is based primarily on regular visual inspection of trees and DNA polymerase chain reaction (PCR) [2]. Visual inspection needs skills to determine HLB infection, while PCR requires specialized laboratory equipment with much cost [3]. Alternative diagnostic measures that are cost-effective, efficient, and highly accurate are urgent issues to develop. Optical image analyses could be a potential candidate for HLB detection [4].

Diagnosing plant diseases with RGB image analyses, particularly with convolutional neural networks (CNNs), has been enthusiastically explored in this decade [4]. These analyses rely on HLB symptoms appearing on trees: yellow blotchy mottling, thickening, and enlarged or corky veins of leaves [5], and wilting, stunted growth, or yellow shoots of the canopy [1]. Kalim et al. [6] achieved 87% precision in distinguishing HLB from other five diseases with combining machine learning and deep learning on 594 leaf images with these symptoms. Elaraby et al. [7] reached 94.3% accuracy in diagnosing six citrus diseases with an SGDM+AlexNet model that used 4745 leaf images, while Gómez-Flores et al. [8] used a VGG19 learning model in which they transferred 953 leaf image data distinguished into HLB-infected and not infected groups. This model correctly identified the HLB infection with 95% precision. Deep learning with numerous RGB images thus achieves high precision in detecting the HLB-infected trees. However, various background environmental noises are also taken in the images and the elimination of such noises negatively interferes with the image analyses [9]. Furthermore, the success in image analyzing depends on how appropriately the targeted samples are photographed, being determined by the experience on visual disease inspection [10].

Finer spectral data with finer division of wavelengths than RGB contribute to successfully excluding environmental noises and provide more effective image data for crop health, pest, and soil assessment in agriculture [4]. This technique utilizes the electromagnetic intensity of spectra that are determined by the chemical and physical properties in plants [11]. Two measures are adopted, depending on the differences in the number of wavelengths: multispectral analyses in which 4–16 wavelengths are used and hyperspectral analyses that the optical data are collected at 200–1000 wavelengths divided at every 1 to 7 nm (Figure 1). Owing to its finer partitioning of wavelengths to analyze, hyperspectral imaging usually gives more accuracy in separating different groups, in particular detecting asymptomatic infection [12]. For example, multispectral analyses with three spectral regions (520–600 nm, 630–690 nm, 760–900 nm) succeed in detecting HLB in an accuracy > 90% with four machine learning techniques on the image data that were obtained from aerial viewing of an orchard by a drone [13]. On the other hand, Deng et al. [14] analyzed hyperspectral data and extracted 13 specific wavelengths using six machine learning models, successfully distinguishing HLB infection with a precision of 96%, being more sensitive by 6% than the former.

Although hyperspectral cameras may specify the disease infection in an early stage, including HLB, they are costly and impractical for large-scale use [15]. Confining the data in wavelengths that contribute to the differentiation in the reflectance properties in HLB-infected trees, the size-down in image data could be possible, and more handy equipment for HLB detection may be able to develop. Various authors have published a number of articles devoted to the determination of HLB using hyperspectral remote sensing and have attempted to extract some characteristic wavelengths (Table 1, [14,16,17,18,19,20,21,22,23,24]). In this paper, we propose statistical methods to efficiently confine useful wavelengths from hyperspectral data and machine learning models to identify HLB infection.

## 2. Results

### 2.1. PCA for the Explanation of the Variance in the Data

The contributions of the first eight principal components (PCs) revealed by the PCA (principal component analysis) were 83.44%, 12.83%, 2.17%, 0.84%, 0.14%, 0.08%, 0.06%, and 0.04%, respectively (Figure 2a). Since the first four PCs explained over 99% of the variance in the data, we output their loading plot as shown in Figure 2b. The loadings represent the contribution of each wavelength to the distinction of image groupings, asymptomatic and symptomatic. Figure 2b shows relatively large loading variations in the 500–600 nm, near 700 nm, and 900–1000 nm regions. This suggests that the reflectance properties of the foliar hyperspectral data vary the most at these wavelengths, which may be critical for HLB detection classification. The loading curve for PC1 was smoother and usually represented the main differences in overall reflection or intensity. PC2–4 explained smaller proportions of the variance and showed more peaks and troughs, often representing minor but still important differences or possibly noise.

We then compared the distribution of asymptomatic and symptomatic data on PC1–4 two by two and output their scatter plots (Figure 3). Regardless of which two PCs are compared, there is a large overlap region, which indicates that PCA only is not yet able to separate the differences between the two categories well. Further wavelength optimization may be needed to exclude some redundant information. We analyzed the top eight wavelengths of PC1–4 and their loadings as shown in Table 2. Wavelengths closely regressed on PC1 in the range 715–736 nm, which corresponded to the red-edge region of citrus leaves. On PC2, the wavelengths at the 930–960 nm range were regressed, expanding over the near-infrared region. Wavelengths on PC3 included shorter wavelengths, 400, 710, and 933 nm, while on PC4 near infrared wavelengths 933–957 nm were related, mostly overlapping with PC2. Since PC1 and PC2 explained more than 96% of the variance and there was no overlap in the top eight wavelengths, a combination of these two PCs was chosen for further wavelength optimization. The PC1 score of the symptomatic group was negatively correlated against the PC2, while asymptotic groups were scattered near the origin in the coordinates with no apparent correlation.

### 2.2. Hyperspectral Data Wavelength Optimization

To further refine HLB characteristic wavelengths, we applied the wavelength optimization method based on incremental feature selection. By sequentially increasing the number of selected wavelengths according to their loading scores, we obtained a total of 64 wavelength combinations. These combinations were then input into the seven models for training and the F1 scores were compared for three conditions: full wavelengths, feature extraction, and wavelength optimization.

Nonlinear models (random forest (RF), decision tree, k-nearest neighbor (KNN), gradient boosting, and support vector machine (SVM)) consistently achieved F1 scores above 95%, while linear models (linear discriminant analysis (LDA) and logistic regression) performed poorly, with F1 scores below 40% after data reduction (Table 3). Among the nonlinear models, only RF, decision tree, KNN, and gradient boosting maintained high F1 scores (>95%) after wavelength optimization. The RF model integrating nine wavelengths achieved the highest F1 score of 99.8%, and the gradient boosting model integrating twelve wavelengths achieved the lowest F1 score of 96.2%. We took these four models and discussed them further in terms of their reliability.

### 2.3. Reliability of Expert System

A PCR-based study relied on precise PCR results to build a hyperspectral dataset of foliar leaves, which was learned and tested using six different machine learning models and achieved 96% accuracy on a 13-wavelength combination. As the four nonlinear models mentioned above performed well after data reduction, we conducted nearest neighbor matching for comparing wavelengths extracted in wavelength optimization by them with this 13-wavelength combination reported in the PCR-based study (Table 4). The discrepancies between PCR-based and model-selected wavelengths were 2.78 nm for RF, 3.14 nm for decision tree, 3.5 nm for KNN, and 5.0 nm for gradient boosting (Figure 4). Owing to their discrepancies smaller than the hyperspectral camera’s resolution of 7 (0 ± 3.5 nm), the RF, decision tree, and KNN models can be taken as effective alternatives to the PCR test for HLB detection. Therefore, these three models were selected for further analyses.

We analyzed the F1 scores, wavelengths used, and test time of these three models (Table 5). The decision tree with seven wavelengths completed the calculation in the shortest time, 7 ms, but the precision was comparable to that of the RF model. Although the KNN model required the longest time in the calculation, it utilized only four wavelengths with keeping the F1-score higher. These findings indicate that the RF, decision tree, and KNN models in which red to infrared reflectance data are integrated can recognize HLB infection with more than 97% precision. In particular, the RF and decision tree models almost perfectly separated image groups.

The availability of models may be considered in three aspects: precision in HLB determination, quickness in calculation time, and the reduction of used wavelengths (Figure 5). If focusing only on the precision, the RF model will be selected. On the other hand, if calculation time is the priority, the decision tree model should be the first choice. If simpler equipment is preferred, the KNN model should be considered because it uses only four wavelengths, which may be easily integrated into the multispectral analyzing system. Since all three models have high-level precision while the decision tree model has an outstanding processing speed, we would recommend prioritizing this model.

Permutation tests were conducted on these three models selected above to validate the statistical significance in their HLB determination. We randomly shuffled the true labels of the test set 10,000 times and evaluated them using the same trained models. All three models were significant (*p* < 0.01), meaning that RF, decision tree, and KNN models all performed HLB detection significantly better than random performance.

## 3. Discussion

The RF, decision tree, and KNN models indicated that red-edge to infrared wavelengths at 727 to 960 nm significantly contributed to determine leaves’ HLB symptoms from symptomatic leaves (Table 4). The 727 nm wavelength lies on the red-edge of visual light, which is typically closely related to the chlorophyll a content in plant leaves and their photosynthetic activities [25,26]. HLB infection disrupts chlorophyll metabolism, leading the optical characteristics at this wavelength to change and, accordingly, other waves around it also may be modified. Wavelengths at 930–960 nm fall within the near-infrared region and are associated with variations in leaf water content and cellular structure [27]. The results in this present study indicate that wavelengths differentiated in these bands may reflect the changes in either or both characters in HLB-infected leaves. The three models selected in our study thus could be integrated into the HLB-infection system as more cost-effective and efficient sensing technologies.

Meanwhile, there are some limitations in our study as well. We chose the study of Deng et al. [14] for comparison because their equipment, imaging conditions, and feature selection methods are similar to ours. This alignment can provide a more meaningful and fair comparison to highlight the reliability and applicability of our approach. However, accurate PCR tests should have been performed on the collected leaf samples to provide a more objective corroboration of the results of the expert inspection. Although our models successfully achieved almost perfect distinction between symptomatic and asymptomatic leaves with the integration of the two major components, in some cases, minor components often make important contributions to category-distinction. Ignorance of such minor components sometimes reduces the performance of models [28]. The existence of a partial overlap in the scatter plot of PC1 and PC2 (Figure 3a) suggests that leaf refraction in some HLB-infected leaves could not be differentiated from non-infected leaves based solely on the wavelengths explained by PC1. This insufficient separation in the HLB-infected leaves may be apparent in the early stages of infection, where few wavelengths lead noticeable changes in this stage. Such incomplete differentiation, if any, would result in the failing in the distinction of HLB infection by these three models. Therefore, the overlap might be attributed to how long the leaf has been infected by HLB: the shorter the time since the HLB infection, the less the wavelength traits, making early differentiation more challenging. Minor components other than PC1 and PC2 may contribute to distinguishing early infected leaves, which deserves further investigation. In addition, a mentioned nonlinear dimensionality reduction technique, t-SNE, has shown better results than PCA in feature extraction of some hyperspectral data [29,30]. We also plan to try to use this method to solve the overlapping problem mentioned above.

Our expert system is trained with images taken from multiple leaves and applied to the real-world recognition. Thus, multiple leaves can be determined into a presumed category at once by taking similar images. Compared to models that require taking pictures of one leaf at a time, the detection efficiency can be improved several times more precisely. Nonetheless, it is still inevitable to reach an incorrect determination or perform the determination less effectively. Recently, eye-tracking studies reveal experts assess HLB by scanning broader areas, suggesting the use of holistic cues not captured by current deep learning models [31]. Given the successful application of attention mechanisms in deep learning models in recent years [32], we propose that integrating expert diagnostic strategies can enhance model performance. By learning from the judgment experience of seasoned experts, deep learning models can establish key attention areas on the whole tree, improving learning efficiency and diagnostic accuracy. This approach has the potential to simplify the HLB diagnosis process and contribute to more effective disease management strategies, which is our future work.

## 4. Materials and Methods

Figure 6 illustrates our data processing flow, where foliar images were obtained with a hyperspectral camera (Specim IQ, SPECIM, Oulu, Finland) of 204 wavelengths partitioned at an interval of about 7 nm at 400–1000 nm. Based on expert visual diagnosis, hyperspectral pixel data were divided into two categories: any symptoms on leaves (symptomatic) or no symptom (asymptomatic). The pixel data were randomly split into training (80%) and test (20%) sets by label proportions. PCA was applied on every individual wavelength to extract specific wavelengths that were characteristic in foliar symptomatic appearances. PCA was applied to the training set only to avoid overfitting due to data leakage [33]. Then, an incremental feature selection method was conducted to explore the possible combinations of the PCA-determined key wavelengths, and the optimal wavelength combinations were determined on each of the seven machine learning models based on the F1-score metric. These models included five nonlinear models: RF, decision tree, SVM, KNN, and gradient boosting, and two linear models: logistic regression and LDA. Finally, nearest neighbor matching and permutation test methods were used to compare the HLB characteristic wavelengths extracted by the expert visual inspection and PCR-based detection to validate the reliability of our expert system.

### 4.1. Hyperspectral Image Data Collection

All images in this study were obtained in a citrus orchard, approximately 490 m^2^ located in Kuet Chang Sub-district, Mae Taeng District, Chiang Mai, Thailand (Figure 7). In this orchard, 7-year-old trees of the local tangerine cultivar, Sai Num Phung (*Citrus reticulata*), were cultivated, spaced 4 m from each other on a row and 3 m between rows.

Hyperspectral images were collected at 10:30 to 11:30 am on a sunny morning on 7 August 2024, with the hyperspectral camera fixed on a tripod. The distance between the front lens and imaged leaves was 40 cm and a neutral density filter (Ni060-40.5, MIDOPT, Palatine, IL, USA) was attached on the top of the objective lens to reduce light intensity. This filter reduced light intensity which would adversely affect image captures. A white reference panel (IQ White Reference, SPECIM, Oulu, Finland) was set beside plants for reflectance calibration every time the plants were imaged (Figure 8). Table 6 shows the devices and their specifications in this study. A total of 22 branch images were collected from 10 trees in this study.

### 4.2. Hyperspectral Data Preprocessing

A python-based preprocessing software developed by us was utilized to eliminate background noises in the images and extract area needed for the analyses using HSV (hue, saturation, value) thresholds and manual selection (Figure 9). This software extracted spectral data of 72 leaves in the 22 branch images. Expert visual assessment identified HLB symptoms in four out of the ten trees, resulting in 12 symptomatic and 60 asymptomatic leaves. All pixels from symptomatic leaves were labeled as “symptomatic” and those from asymptomatic leaves as “asymptomatic”. We eventually extracted 63,056 “symptomatic” pixels and 294,886 “asymptomatic” pixels. Considering the sample imbalance, we used stratified sampling in dividing the dataset, 80% from each of “symptomatic” and “asymptomatic” as the training set and the remaining 20% as the test set.

### 4.3. Hyperspectral Data Analysis and Modeling

#### 4.3.1. Feature Extraction—PCA

PCA is a statistical technique used to simplify complex datasets while retaining as much important information as possible, widely used in fields like image processing, genetics, and finance to reduce data complexity and uncover meaningful patterns [34]. In this study, it was used to focus on wavelengths that could have differentiated optical characteristics of leaf reflectance due to the HLB symptoms. The PCs were created and calculated, and the proportion of variance explained by each component was used as a criterion for selecting key features. Often, just a few of these new components capture most of the information in the dataset. By focusing on these few components, the dataset can be simplified while preserving its essence. Before performing PCA, all spectral data were normalized to ensure uniform scaling across wavelengths.

#### 4.3.2. Wavelength Optimization—Incremental Feature Selection

Since 16 spectral are the maximum wavelengths available for multispectral cameras, the extraction of wavelengths in the hyperspectral data was limited to or fewer than 16 wavelengths. However, 16 wavelengths rise to 2^16^ − 1 = 65,535 combinations, except for the null combination, which impose heavy computational burden. In this study, this problem was avoided by adopting an incremental feature selection approach to generate targeted wavelength combinations (Table 7). This approach aligns with the data-driven insights provided by PCA, focusing on the most informative wavelengths from PCs rather than all available features [35]. Specifically, this method systematically explores combinations by gradually increasing the number of wavelengths from each component, ensuring a balanced representation of features while maintaining a logical structure.

#### 4.3.3. Machine Learning Models

Seven machine learning models were assessed for how precisely they distinguished images in two groups, symptomatic or asymptomatic. Models examined in this study were five nonlinear models (RF [36], decision tree [37], KNN [38], gradient boosting [39], and SVM [40]) and two linear models (logistic regression [41] and LDA [42]). These methods are well-established in the current machine learning and were chosen due to their classification capabilities [14].

### 4.4. Evaluation of Models for Leaf Image Separation

#### 4.4.1. F1 Score

F1 score was used to evaluate the precision of machine learning models in separating image data into the groups. This is a powerful tool for imbalanced datasets, as it adopts the harmonic mean of precision and recall, and effectively regulates both false positives and false negatives to provide a comprehensive performance measure [43]. The F1 score is calculated through the following three equations:(1)$$Precision\;(\%)=\frac{TP}{TP+FP}\times 100$$(2)$$Recall\;(\%)=\frac{TP}{TP+FN}\times 100$$(3)$$F1-score\;(\%)=\frac{2\times Precision\times Recall}{Precision+Recall}\times 100$$
where TPs (True Positives) means correctly identified positive cases; TNs (True Negatives) means correctly identified negative cases; FPs (False Positives) means negative cases incorrectly identified as positive; FNs (False Negatives) means positive cases incorrectly identified as negative [44].

#### 4.4.2. Nearest Neighbor Matching

Nearest neighbor matching is a statistical technique commonly used in observational studies to pair each unit in a treatment group with the closest unit in a control group based on a predefined set of covariates or characteristics [45]. Given two wavelength sets, say A = {*a*_1_, *a*_2_, … *a_n_*} and B = {*b*_1_, *b*_2_, …, *b_m_*}, the matching calculates the mean of the differences between each wavelength in A and its nearest wavelength in B as a measure of similarity:(4)$$Distance=\frac{1}{n}\sum_{i=1}^{n}\left(\min\left|a_{i}-b_{j}\right|\right)\;for\;each\;a_{i}\in A,b_{j}\in B$$

To validate the effectiveness of our expert’s visual inspection, we used nearest neighbor matching to compare the HLB characteristic wavelengths extracted by our models with those reported in a PCR-based study. That study utilized hyperspectral imaging based on PCR testing to analyze the optimal wavelength combinations, and 13 HLB characteristic wavelengths (544, 718, 753, 760, 764, 930, 938, 943, 951, 969, 985, 998, and 999 nm) achieved the best performance of an accuracy of 96% [14]. Using nearest neighbor matching, we calculated the distance errors between the wavelengths extracted by each model and those from the PCR-based study.

#### 4.4.3. Permutation Test

The permutation test is a robust, non-parametric statistical method which assesses whether the performance of a machine learning model is significantly better than random performance [46]. The procedure involves shuffling target labels to break the relationship between the input feature and the target variable, followed by re-evaluating the model on the permuted dataset multiple times (at least 1000 times is required for reliability) [46]. In this study, we randomly shuffled the true labels of the test set and evaluated the shuffled labels using the same trained models. This process was repeated 10,000 times. The frequencies of F1 scores greater than or equal to the scores achieved with the true labels were counted and the model’s statistical significance in the HLB determination was examined at *p* < 0.01 [46].

## 5. Conclusions

The following conclusions can be drawn from the study:-Our RF, decision tree, and KNN models are as reliable as PCR in identifying HLB.-Nonlinear models outperform linear models for HLB spectral data.-Using PCA for nonlinear models is effective for HLB feature extraction.-Decision tree model provides high accuracy with faster prediction, suitable for real-time applications.-KNN model shows promising potential for multispectral imaging applications.-The red-edge and near-infrared regions may be critical for HLB detection.

Future work will focus on applying these findings to multispectral cameras and analyzing expert visual patterns to develop deep learning models for whole-tree HLB diagnosis.

## Figures and Tables

**Figure 1 plants-14-00451-f001:**
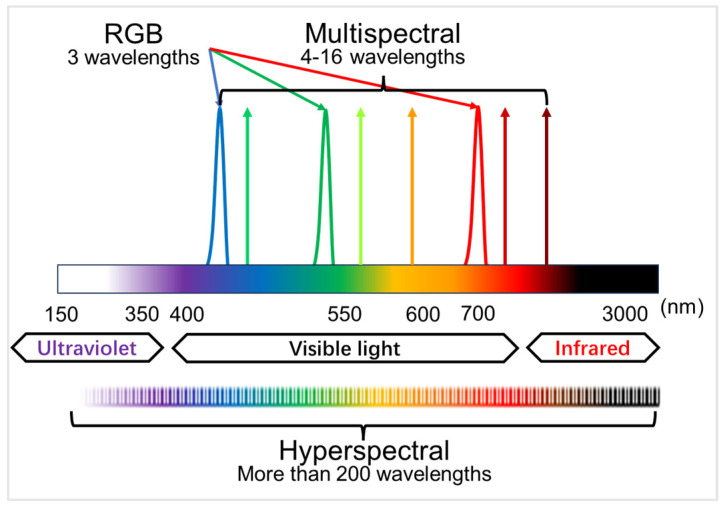
The schematic of RGB, multispectral, and hyperspectral images.

**Figure 2 plants-14-00451-f002:**
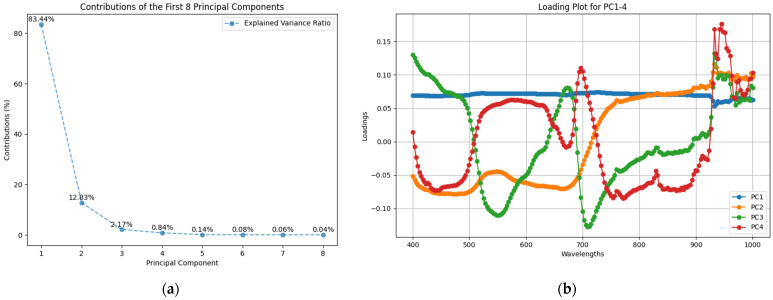
The contributions of the first eight principal components (**a**) and the loading plot for principal components 1–4 (**b**).

**Figure 3 plants-14-00451-f003:**
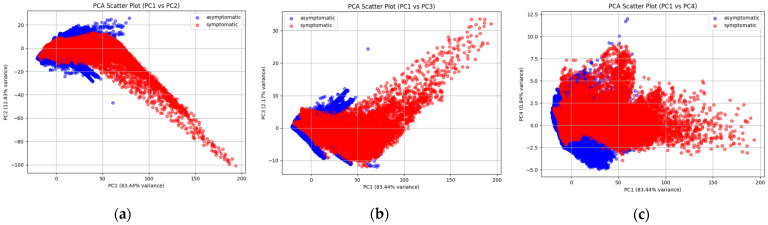
The scatter plots of different PCs between PC1 and PC4 (**a**–**f**).

**Figure 4 plants-14-00451-f004:**
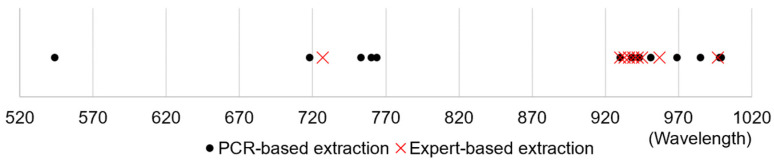
Comparison of key wavelengths identified by the PCR-based study and our expert-based RF model.

**Figure 5 plants-14-00451-f005:**
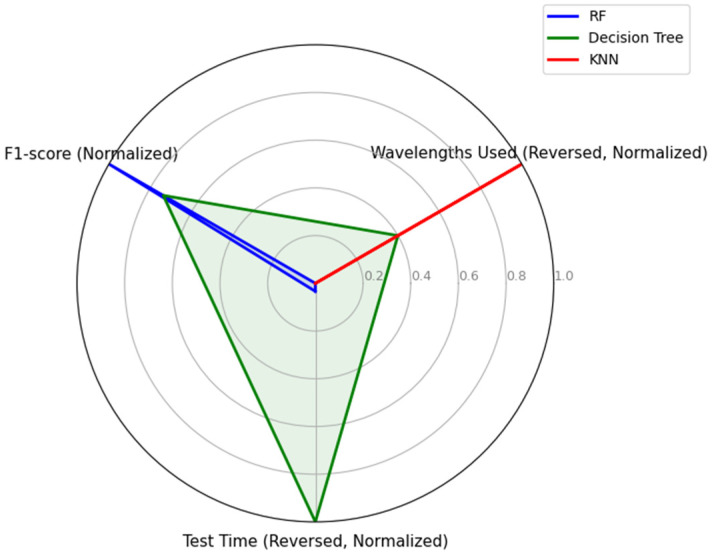
Normalized performance comparison of classification models with F1 score, reversed test time, and reversed wavelengths used.

**Figure 6 plants-14-00451-f006:**
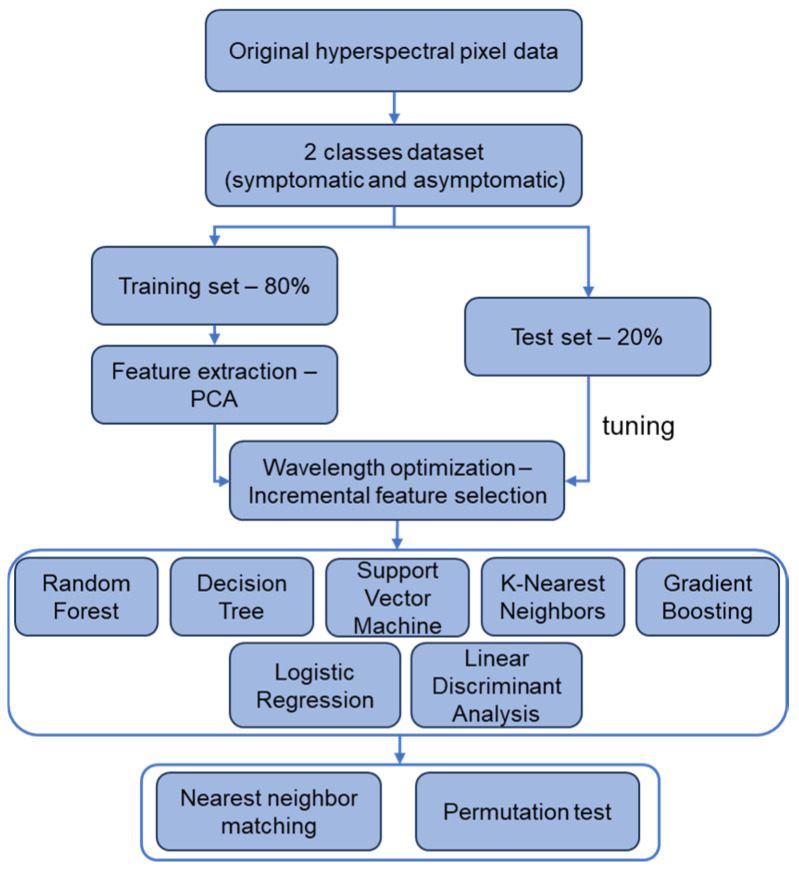
The flowchart of hyperspectral data analysis and modeling in this study.

**Figure 7 plants-14-00451-f007:**
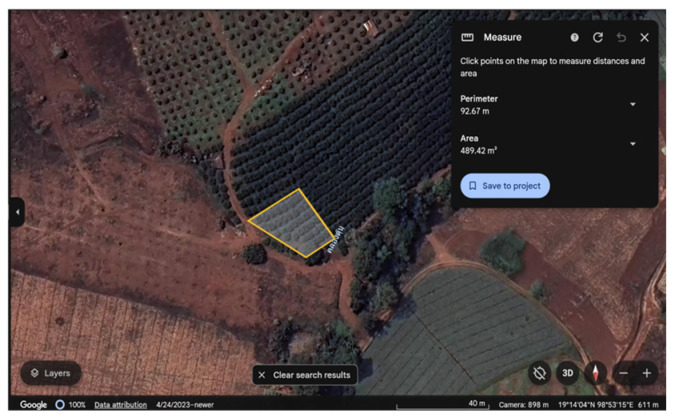
The orchard in Thailand for this study (19°14′02″ N 98°53′14″ E, 611 m AMSL). This image is available from Google Earth. The Thai word in the figure is the local place name.

**Figure 8 plants-14-00451-f008:**
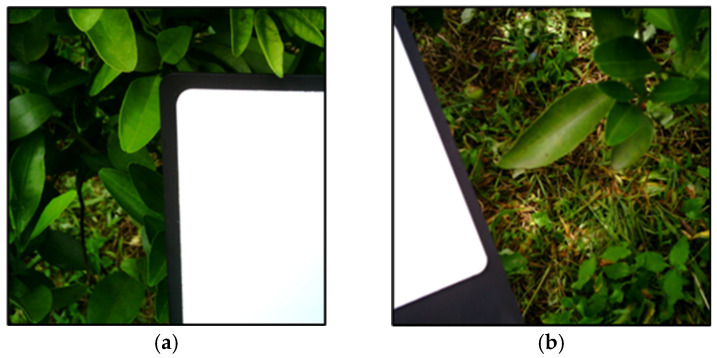
RGB images of branches in symptomatic (**a**) and asymptomatic (**b**) groups created from the composite of hyperspectral images.

**Figure 9 plants-14-00451-f009:**
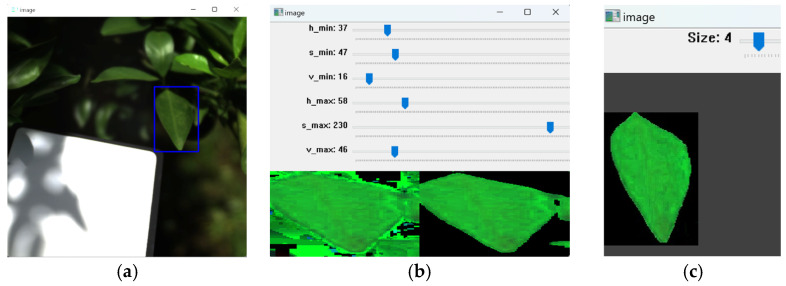
Schematic of our preprocessing software. First, the region of interest is manually selected (blue region) (**a**). Background noise is then largely removed by HSV thresholding (**b**), with manual adjustment for finer noise removal (**c**). The blue boxes can be used to control the threshold in (**b**) and the size of the pixels when manually removing in (**c**). Finally, hyperspectral data of the selected region are extracted and saved.

**Table 1 plants-14-00451-t001:** Previous studies using hyperspectral remote sensing for HLB detection.

Publication Year	Equipment Wavelengths (nm)	Feature Wavelengths (nm)	Study Type
2011–2013 [16,17,18]	350–2500	537, 612, 638, 662, 688, 713, 763, 813, 998, 1066, 1120, 1148, 1296, 1445, 1472, 1546, 1597, 1622, 1746, 1898, 2121, 2172, 2348, 2471, 2493	field
2012 [19]	457–921	650–850	field and lab
2012 [20]	457–921	410–432, 440–509, 634–686, 734–927, 932, 951, 975, 980	field
2018 [21]	379–1023	493, 515, 665, 716, 739	lab
2019 [14]	400–1000	544, 718, 753, 760, 764, 930, 938, 943, 951, 969, 985, 998, 999	field
2020 [22]	450–950, 325–1075	468, 504, 512, 516, 528, 536, 632, 680, 688, 852	field
2024 [23]	400–1000	560, 678, 726, 750	lab
2025 [24]	325–1075	375–425, 650–750, 890–925	lab

**Table 2 plants-14-00451-t002:** Top eight wavelengths of principal components 1–4 and their loadings.

PC1		PC2		PC3		PC4	
Wavelength	Loading	Wavelength	Loading	Wavelength	Loading	Wavelength	Loading
727	0.0741	933	0.1165	933	0.1324	945	0.1764
724	0.0741	936	0.1111	400	0.1303	942	0.1686
730	0.0740	930	0.1040	709	−0.1276	933	0.1683
721	0.0739	957	0.1032	712	−0.1271	948	0.1648
733	0.0739	942	0.1031	936	0.1260	951	0.1636
718	0.0738	939	0.1025	403	0.1254	954	0.1400
736	0.0737	945	0.1025	706	−0.1247	957	0.1359
715	0.0735	997	0.1025	715	−0.1232	936	0.1312

**Table 3 plants-14-00451-t003:** F1 scores of seven classification models with symptomatic and asymptomatic datasets for full wavelengths, feature extraction, and wavelength optimization.

Classification Model		Full Wavelengths (204 Wavelengths)	Feature Extraction (16 Wavelengths)	Wavelength Optimization (No. of Wavelengths)
Nonlinear	RF	99.5%	99.6%	99.8% (9)
	Decision tree	99.1%	99.2%	99.3% (6)
	KNN	98.8%	96.3%	97.9% (4)
	Gradient boosting	97.7%	95.6%	96.2% (12)
	SVM	98.4%	89.2%	89.2% (16)
Linear	LDA	89.3%	38.8%	38.8% (15)
	Logistic regression	91.8%	38.3%	38.4% (14)

**Table 4 plants-14-00451-t004:** Extracted wavelengths by RF, decision tree, KNN, and gradient boosting models and their wavelength discrepancies between the study by Deng et al. [14].

Classification Model	Extracted Wavelengths (nm)	Discrepancies
Deng et al. [14]	544, 718, 753, 760, 764, 930, 938, 943, 951, 969, 985, 998, 999	
RF	727, 930, 933, 936, 939, 942, 945, 957, 997	2.78
Decision tree	727, 930, 933, 936, 939, 942, 957	3.14
KNN	727, 930, 933, 936	3.5
Gradient boosting	721, 724, 727, 730, 733, 930, 933, 936, 939, 942, 945, 957	5.0

**Table 5 plants-14-00451-t005:** The F1-scores, wavelengths used and test time achieved by RF, decision tree and KNN models.

Classification Model	F1-Score	Wavelengths Used	Test Time (ms)
RF	99.8%	9	869
Decision tree	99.3%	7	7
KNN	97.9%	4	899

**Table 6 plants-14-00451-t006:** Specification of the hyperspectral camera used in this study.

Device	Specification	Value
Specim IQ	Resolution	512 × 512 pix
	Wavelength range (204)	397–1004 nm
	Dimension	207 × 91 × 74 mm
	Pixel size	17.58 μm × 17.58 μm
Calibration whiteboard	Reflectivity	100%
	Size	10 × 10 cm
Neutral density filter	Average Transmission	25%

**Table 7 plants-14-00451-t007:** An example of an incremental feature selection approach focusing on the first two components, the primary and secondary principal components, PC1 and PC2, respectively.

PC1	PC2	Wavelength Selection	Counts
*A* _1_	*B* _1_	*A* _1_ *B* _1_ *, A* _1_ *B* _1_ *B* _2_ *, …, A* _1_ *B* _1_ *B* _2_ *B* _3_ *… B* _n_	*n*
*A* _2_	*B* _2_	*A* _1_ *A* _2_ *B* _1_ *, A* _1_ *A* _2_ *B* _1_ *B* _2_ *, …, A* _1_ *A* _2_ *B* _1_ *B* _2_ *B* _3_ *… B* _n_	*n*
*A* _3_	*B* _3_	*A* _1_ *A* _2_ *A* _3_ *B* _1_ *, A* _1_ *A* _2_ *A* _3_ *B* _1_ *B* _2_ *, …, A* _1_ *A* _2_ *A* _3_ *B* _1_ *B* _2_ *B* _3_ *… B* _n_	*n*
*……*		*……*	*n*
*A* _n_	*B* _n_	*A* _1_ *A* _2_ *A* _3_ *… A* _n_ *B* _1_ *, A* _1_ *A* _2_ *A* _3_ *… A* _n_ *B* _1_ *B* _2_ *, …, A* _1_ *A* _2_ *A* _3_ *… A* _n_ *B* _1_ *B* _2_ *B* _3_ *… B* _n_	*n*

## Data Availability

The data presented in this study are available on request from the corresponding author. Software: PyCharm Community Edition 2024.2.1 for hyperspectral data analysis and machine learning modeling; Anaconda3 for managing library files.

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
