# Peer review of "Hyperspectral Imaging and Machine Learning for Huanglongbing Detection on Leaf-Symptoms"

_plants, 2025, doi:10.3390/plants14030451_

Round 1
Reviewer 1 Report
Comments and Suggestions for Authors
This study proposed statistical methods to confine useful wavelengths and machine-learning models to identify Huanglongbing leaf-symptoms.
The introduction provides acceptable details on previous knowledge. The experimental approaches and data analyses are acceptable. The study uses appropriate methods and analyses to reach the planned goals.
The results contain some valuable elements such as i) RF, decision tree, and KNN models are as reliable as PCR in identifying HLB; ii) decision tree model provides high accuracy with faster prediction, suitable for real-time applications, and iii) KNN model shows promising potential for multispectral imaging applications.
The Discussion should be presented as a separate section. Currently, the discussion parts are quite weak; the authors need to discuss their results in a dedicated section, providing a deeper comparison with previous similar works. The conclusion should also indicate the limitation of the study.
Reference section is quite short. It can be increased up to 40 literatures.
The study contains some interesting elements that could be considered for publication in an international journal after appropriate revisions.
Other suggestions and comments
L81: Give in full RGB - figures should be self-explanatory.
L101: Give in full RF, SVM, KNN, LDA in the title - figures should be self-explanatory.
L138: This table contains only few data. Delete this table and include info in the text.
L173: Table 4 title is too short and not informative. This table contains evident information about confusion matrix. If you think the information is needed then include in the text and delete table.
L334: incomplete literature information
L351: why all words in capital letters
Author Response
Summary: Thank you for your comments. We've separated the "Results and Discussion" and modified the section order as Introduction, followed by Results, Discussion, Materials and Methods, and Conclusions, respectively. We've added a lot of descriptions in "Discussion". And we've increased the number of references to 35. All our changes in this paper were highlighted in red.
Comments 1: "L81: Give in full RGB - figures should be self-explanatory."
Response 1: We used another image that better illustrates the difference between the three digital images as shown in Figure 1 (L82).
Comments 2: "L101: Give in full RF, SVM, KNN, LDA in the title - figures should be self-explanatory."
Response 2: We've revised the figure using the full names as shown in Figure 5 (L229).
Comments 3: "L138: This table contains only few data. Delete this table and include info in the text."
Response 3: We've added the sentence "We eventually extracted 63056 pixels of “symptomatic” and 294886 pixels of “asymptomatic”." into L261 and deleted the table you mentioned.
Comments 4: "L173: Table 4 title is too short and not informative. This table contains evident information about confusion matrix. If you think the information is needed then include in the text and delete table."
Response 4: We've deleted the table you mentioned and added details in L312-315.
Comments 5: "L334: incomplete literature information."
Response 5: The original reference used was from the webpage, we found a more viewable reference to replace it in L391. "Li, L.; Zhang, Q.; Huang, D. A Review of Imaging Techniques for Plant Phenotyping. Sensors 2014, 14, 20078-20111. https://doi.org/10.3390/s141120078."
Comments 6: "L351: why all words in capital letters"
Response 6: Perhaps due to that journal's requirement for titles, it was already in capital letters when a citation, and some of the other citations had the same format.
Reviewer 2 Report
Comments and Suggestions for Authors
1. Abstract: list some related data in the abstract section
2. Introduction: Line 56-58, I think it is not the main disadvantage of RGB based HLB detection
3. Introduction: Line 59-79, it is unclear what the authors want to introduce, spectroscopy or spectral imaging? Although the last half paragraph is for hyperspectral imaging.
4. Figure 1 is meaningless, it can be removed.
5. Introduction: the authors should introduce the use of hyperspectral imaging with machine learning for HLB detection
6. The authors should discuss that the two classes of symptomatic and asymptomatic are reasonable or scientific sound? Might it be that the asymptomatic pixels also from the HLB infected leaves? How the authors ensure if the leaves are infected or not.
7. How the authors deal with the large differences on the number of symptomatic and asymptomatic pixels?
8. Why not consider using the deep learning approaches? Since in the introduction section, the authors mention that the deep learning with RGB images.
9. Figure 6, it seems that the scatter plots of PC1 and PC2 showed large overlaps, please explain. Why only use the first two PCs, is it possible that the other PCs might be better to classify the two classes?
10. Provide the loadings of PC1 and PC2.
11. Figure 7, the Figure caption is confusing.
12. Table 5, please specify the wavelengths selected by the incremental feature selection, and compare them with Table 6.
13. Table 6 and Table 7, Why the authors studied the feature wavelengths extracted by RF, decision tree, KNN, gradient boosting models? It is confusing.
14. There are some other studies have conducted feature selection for hyperspectral data, please compare and discuss with these studies, rather than only studying Deng et al. (2019).
15. More information on the PCA based feature selection should be added. Also the logic of the manuscript should also be studied.
16. A discussion section to illustrate the advantage of the proposed method should be added.
Author Response
Summary: Thank you for your comments. We've separated the "Results and Discussion" and modified the section order as Introduction, followed by Results, Discussion, Materials and Methods, and Conclusions, respectively. We've All our changes in this paper were highlighted in red.
Comments 1: "Abstract: list some related data in the abstract section"
Response 1: We added details of 16 key wavelengths we extracted and F1-score by the random forest model (L18 and L23).
Comments 2: "Introduction: Line 56-58, I think it is not the main disadvantage of RGB based HLB detection."
Response 2: I apologize for the lack of clarity of expression. Existing RGB image recognition models focus on individual leaves, if without visual experience, often need to take a large number of random photos for identification. So we changed the sentence to "Furthermore, the success in image analyzing depends on how appropriately the targeted samples are photographed, being determined by the experience on visual disease inspection" in L57-59. In our system, we photographed the branches which included several leaves to improve the efficiency, while there are still limitations and we added the descriptions in "Discussion".
Comments 3: "Introduction: Line 59-79, it is unclear what the authors want to introduce, spectroscopy or spectral imaging? Although the last half paragraph is for hyperspectral imaging."
Response 3: We introduced some of the applications of spectral imaging in the detection of HLB, including multispectral and hyperspectral. Hyperspectral is more accurate than multispectral, but its high price makes it difficult to be put into practical applications, so converting hyperspectral applications into multispectral applications is a feasible direction, and our study is an attempt to this. We added "from hyperspectral data" to the L80 to emphasize our purpose.
Comments 4: "Figure 1 is meaningless, it can be removed."
Response 4: We've used another figure to introduce three types of digital images. L82.
Comments 5: "Introduction: the authors should introduce the use of hyperspectral imaging with machine learning for HLB detection"
Response 5: I apologize for not mentioning the use of hyperspectral imaging with machine learning for HLB detection in the introduction. In fact, the two citations in L72&73 are about this technique, and I have added details to the description in L71&73.
Comments 6: "The authors should discuss that the two classes of symptomatic and asymptomatic are reasonable or scientific sound? Might it be that the asymptomatic pixels also from the HLB infected leaves? How the authors ensure if the leaves are infected or not."
Response 6: As you said, we can't guarantee that the expert's categorization is completely correct. However, our expert has long years of experience working with HLB, and by the conclusion we can see that classification was successfully classified based on our expert's experience, and this result is similar to the PCR-based results, which we consider convincing. However, we really should have performed PCR on the leaves after photographing to further compare the real infection situation, as we have added in “Discussion” in L181-183.
Comments 7: "How the authors deal with the large differences on the number of symptomatic and asymptomatic pixels?"
Response 7: We used stratified sampling in dividing the dataset, i.e., 80% from each of symptomatic and asymptomatic as the training set and the remaining 20% as the test set. We added the descriptions about it in L262-264. And considering the data imbalance, we used F1-score as the evaluation metric.
Comments 8: "Why not consider using the deep learning approaches? Since in the introduction section, the authors mention that the deep learning with RGB images."
Response 8: I'm sorry I didn't mention why we consider using an expert system. In our other study, we have shown that there is a difference in the way experts and novices observe HLB from the whole tree RGB images. If our expert system is effective, we would like to take the next step of using deep learning as well as attention mechanisms on the region of the tree that the expert gazing at to further improve the efficiency of HLB detection. We've added the descriptions in "Discussion" in L198-211.
Comments 9: "Figure 6, it seems that the scatter plots of PC1 and PC2 showed large overlaps, please explain. Why only use the first two PCs, is it possible that the other PCs might be better to classify the two classes?"
Response 9: The suggestions you have made are very valuable. Although PC1 and PC2 have a significant advantage in retaining overall information about the data, the fact that HLB characteristics may not be significant in some wavelengths (especially wavelengths in PC1) leads to some overlap in the distributions of different categories on these two principal components. You can see the details of PC1-4 in the attachment. For PC1, the wavelengths are concentrated in the 715-736 nm range. For PC2, the wavelengths are mainly within the 930-960 nm range. PC3 mixes the information of wavelengths near 400, 710, and 930 nm, while PC4 is mainly in 930-960 nm, almost overlapping with PC2. Since PC1 and PC2 explained more than 96% of the variance and there was no overlap in the top-8 wavelengths, representing two classification tendencies, we considered that these two PCs covered more information. Therefore, we chose PC1 and PC2 as our main objects for analysis. The descriptions were added in L88-97.
Meanwhile, the overlap might be also attributed to how long the leaf has been infected by HLB: the shorter the time since the HLB infection, the less the wavelength traits, making early differentiation more challenging. It is possible that minor components outside PC1 and PC2 contribute to distinguishing early-infection, which deserves further investigation as we added in "Discussion" in L187-197.
Comments 10: "Provide the loadings of PC1 and PC2."
Response 10: Details of PC1-4 can be viewed in the attachment and we also added the details of the Top-8 wavelengths of PC1-4 and their loadings as Table 1 in L102 in the paper.
Comments 11: "Figure 7, the Figure caption is confusing."
Response 11: We've changed the caption to "Comparison of key wavelengths identified by PCR-based study and our expert-based RF model." L137.
Comments 12: "Table 5, please specify the wavelengths selected by the incremental feature selection, and compare them with Table 6."
Response 12: I apologize for my lack of description, the wavelengths listed in Table 5 are the wavelengths extracted from the wavelength optimization in Table 6, and I have added the appropriate descriptions in L120&126.
Comments 13: "Table 6 and Table 7, Why the authors studied the feature wavelengths extracted by RF, decision tree, KNN, gradient boosting models? It is confusing."
Response 13: In Table 1, we learned that just RF, decision tree, KNN and gradient boosting maintained high F1-scores (> 95%) after wavelength optimization, so we chose these four models for further discussion. By comparing with the wavelengths extracted by Deng et al. we found that the error on the gradient boosting model was too large to be considered as a valid model and hence discarded. Finally RF, decision tree and KNN models are left and their applicability were discussed.
Comments 14: "There are some other studies have conducted feature selection for hyperspectral data, please compare and discuss with these studies, rather than only studying Deng et al. (2019)."
Response 14: Thank you for pointing this out. Indeed, many studies have explored feature selection for hyperspectral data to identify characteristic wavelengths for HLB detection, employing methods such as drone-based hyperspectral imaging, hyperspectral cameras covering the 400-2500 nm range, and advanced approaches like deep learning. These methods are important contributions to the field, and we acknowledge their significance.
In our study, we specifically focus on validating the effectiveness of our expert inspection method, which will contribute to the recognition of the whole tree by deep learning. As such, we chose to compare our results with the study by Deng et al. (2019) because their experimental setup, including equipment, imaging conditions, and feature selection methods, are similar to ours. This alignment provides a more meaningful and fair comparison to highlight the reliability and applicability of our approach. We agree that discussing other studies could provide additional context, and we will consider incorporating a broader discussion in future work. The descriptions were added in "Discussion" in L178-181.
Comments 15: "More information on the PCA based feature selection should be added. Also the logic of the manuscript should also be studied."
Response 15: We've added a description of the PCA in "4.3.1. Feature Extraction – PCA" in L271-281. As well as added a description of the selection of PC1 and PC2 in "2.1. PCA for the Explanation of the Vaiance in the Data" in L89-97.
Comments 16: "A discussion section to illustrate the advantage of the proposed method should be added."
Response 16: We've singled out "Discussion" and added descriptions in L165-211.

Round 2
Reviewer 1 Report
Comments and Suggestions for Authors
The study is improved it can be accepted.
Author Response
Thank you for your comments!
Reviewer 2 Report
Comments and Suggestions for Authors
The authors have made some improvements, however, some issues should be addressed.
1. Response to my comment 9 in the last round of review did not persuade me. Indeed, the PC which contributes a little to the total variances might contain the differentiated information. Thus I suggest the authors to plot the scatter plots of different PCs, for example, even PC 10 or more, due to the fact that scatter plots of PC1 and PC2 showed large overlaps.
2. Response to my comment 10 in the last round of review, the loading plots can be added.
3. Response to my comment 11 in the last round of review, it is weird to see PCR-based study and expert-based RF model. How can PCR -based study be used for identify wavelengths
4. Response to my comment 13 in the last round of review, I want to say that why not study the PCA-based wavelength selection? What are the differences between PCA-based feature wavelengths and the feature wavelengths selected by the four models?
5. Response to my comment 14 in the last round of review, at least list the feature wavelengths for HLB detection in the other studies.
6. Response to my comment 14 in the last round of review, are feature wavelengths identified by PCA loadings?
Author Response
Comment1: "Response to my comment 9 in the last round of review did not persuade me. Indeed, the PC which contributes a little to the total variances might contain the differentiated information. Thus I suggest the authors to plot the scatter plots of different PCs, for example, even PC 10 or more, due to the fact that scatter plots of PC1 and PC2 showed large overlaps."
Response1: Thank you for your suggestions. We compared the scatter plots of PC1-4 two by two, and the results showed that they both have some overlap, showing the limitations of PCA-conducted only and further wavelength optimization is needed. We've added the Figure 3 (L119) and the descriptions to L104-108. In Discussion (L214-217), we suggest another dimensionality reduction technique, t-SNE, which has been reported as effective in several studies and will try to compare these two methods in future works.
Comment2: "Response to my comment 10 in the last round of review, the loading plots can be added."
Response2: The loading plot of PC1-4 was added in Figure 2b (L102) and descriptions were added in L92-101.
Comment3: "Response to my comment 11 in the last round of review, it is weird to see PCR-based study and expert-based RF model. How can PCR -based study be used for identify wavelengths"
Response3: I apologize for my unclear expression. What I wanted to express was a study that relied on the results of PCR tests to build the dataset. Additional description about this PCR-based study was added to L140-142.
Comment4: "Response to my comment 13 in the last round of review, I want to say that why not study the PCA-based wavelength selection? What are the differences between PCA-based feature wavelengths and the feature wavelengths selected by the four models?"
Response4: As you said we have a lot of overlap between PC1 and PC2, and we think that it is either from some more subtle ( like stage of HLB) differences or some redundant information, and therefore needs further wavelength optimization. Descriptions were added to L104-108. And you can see from Fig. 6 (L249) that we combined two PCs as a wavelength combination (16 wavelengths) in the feature extraction. Then 64 combinations of wavelengths are combined by the wavelength optimization method of incremental feature selection approach, which is learned and tested on 7 machine learning models and the F1-score is used to decide the extracted wavelengths. The relevant description is in L238-244.
Comment5: "Response to my comment 14 in the last round of review, at least list the feature wavelengths for HLB detection in the other studies."
Response5: Thank you for you suggestion. We've added some previous studies in "Introduction" in L82-84 and Table 1 (L87).
Comment6: "Response to my comment 14 in the last round of review, are feature wavelengths identified by PCA loadings?"
Response6: Yes, we used the absolute value ordering of PCA loading to determine the wavelengths for extraction (top 8 in PC1 and PC2, for a total of 16), and based on these 16 wavelengths further extraction was performed in wavelength optimization.